# Homogenizing Non-IID Datasets via In-Distribution Knowledge Distillation for Decentralized Learning

**Deepak Ravikumar**                                                   *dravikum@purdue.edu*
*Elmore Family School of Electrical and Computer Engineering,*
*Purdue University, West Lafayette, IN 47907, USA*

**Gobinda Saha**                                                       *gsaha@purdue.edu*
*Elmore Family School of Electrical and Computer Engineering,*
*Purdue University, West Lafayette, IN 47907, USA*

**Sai Aparna Aketi**                                                   *saketi@purdue.edu*
*Elmore Family School of Electrical and Computer Engineering,*
*Purdue University, West Lafayette, IN 47907, USA*

**Kaushik Roy**                                                        *kaushik@purdue.edu*
*Elmore Family School of Electrical and Computer Engineering,*
*Purdue University, West Lafayette, IN 47907, USA*

**Reviewed on OpenReview:** *https://openreview.net/forum?id=CuyJkNjIVd*

## Abstract

Decentralized learning enables server-less training of deep neural networks (DNNs) in a distributed manner on multiple nodes. One of the key challenges with decentralized learning is heterogeneity in the data distribution across the nodes. Data heterogeneity results in slow and unstable global convergence and therefore poor generalization performance. In this paper, we propose In-Distribution Knowledge Distillation (IDKD) to address the challenge of heterogeneous data distribution. The goal of IDKD is to homogenize the data distribution across the nodes. While such data homogenization can be achieved by exchanging data among the nodes sacrificing privacy, IDKD achieves the same objective using a common public dataset across nodes without breaking the privacy constraint. This public dataset is different from the training dataset and is used to distill the knowledge from each node and communicate it to its neighbors through generated labels. With traditional knowledge distillation, the generalization of the distilled model is reduced due to misalignment between the private and public data distribution. Thus, we introduce an Out-of-Distribution (OoD) detector at each node to label a subset of the public dataset that maps close to the local training data distribution. Our experiments on multiple image classification datasets and graph topologies show that the proposed IDKD scheme is more effective than traditional knowledge distillation and achieves state-of-the-art generalization performance on heterogeneously distributed data with minimal communication overhead[1].

## 1 Introduction

There has been an explosion of Internet of Things (IoT) devices and smartphones around the world (Lim et al., 2020). These edge devices are being equipped with advanced sensors, computing, and communication capabilities. The variety and the wealth of data collected by these devices is opening new possibilities in medical applications (Pryss et al., 2015), air quality sensing (Ganti et al., 2011), and more. As more data

---

[1]Code available at `https://github.com/DeepakTatachar/IDKD`

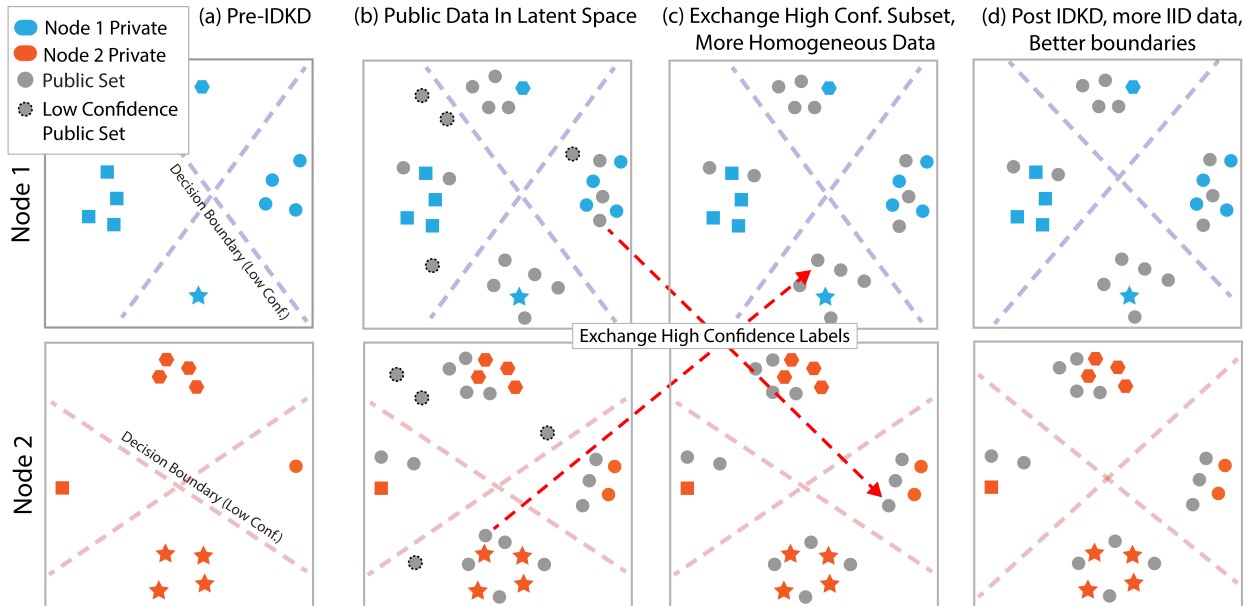

Figure 1: A conceptual overview of IDKD homogenization. We visualize an exaggerated latent space of a 2-node decentralized setup with non-IID data having 4 classes. (a) Two similar networks with small differences in decision boundaries due to data skew. (b) Public data visualized in the latent space (c) Exchange IDKD identified subset using labels only (d) More homogeneous data distribution resulting in better decision boundaries post IDKD training.

is collected at the edge there is a need to not only learn at the edge for efficiency reasons but also maintain privacy (Lim et al., 2020; Wang et al., 2020).

Decentralized deep learning aims to address both these issues by distributing the training process among many compute nodes (Kempe et al., 2003; Tsitsiklis, 1984). Each node works on a subset of the data allowing the use of large models and huge datasets. These local datasets are called private sets as they are not shared with the other nodes. During training, each node communicates gradients or model parameters with its neighbors. The choice of information (gradient, model parameter, etc.) to communicate is dependent on the algorithm used. In general, each node aims to reach a global consensus model using its own local model and updates from its neighbors.

Training DNNs in a decentralized setup gives rise to a number of challenges. The main challenges come from the slow spread of information and when data is distributed in a heterogeneous/non-IID manner across the nodes (Lin et al., 2021). This results in slow and unstable global convergence and poor generalization performance (Lin et al., 2021). To address these issues many methods (Dandi et al., 2022; Duchi et al., 2011; Lian et al., 2017; 2018; Lin et al., 2021; Neglia et al., 2020; Tang et al., 2018; Vogels et al., 2021) have been proposed. Techniques have been proposed to improve the properties of the mixing matrix (Assran et al., 2019; Duchi et al., 2011; Nedić et al., 2018; Neglia et al., 2020) to address the slow spread of information, while most other approaches have primarily focused on the design of decentralized optimization algorithm to handle heterogeneous data distribution (Dandi et al., 2022; Lian et al., 2017; 2018; Lin et al., 2021; Tang et al., 2018; Vogels et al., 2021). In contrast to previous techniques, we aim to homogenize a heterogeneous data distribution without sharing data among nodes to leverage the full potential of existing decentralized algorithms. To achieve this we leverage distillation (Hinton et al., 2015). Distillation has been used to address non-IID distribution in federated learning (FL). However, FL techniques such as DS-FL (Itahara et al., 2021), FedGen (Zhu et al., 2021b), and FedDF (Lin et al., 2020) rely on a central server and thus cannot be readily adopted in a decentralized setting. In Def-KT (Li et al., 2021) the authors proposed a peer-peer knowledge transfer technique however, their algorithm implicitly assumes complete graph connectivity and thus cannot be applied in all decentralized setups. As far as we know our work is the only work that

addresses non-IID distribution from the data perspective in a decentralized setting. Table 1 summarizes previous approaches addressing non-IID data distribution in centralized and decentralized learning.

| Technique | Centralized | Decentralized | Knowledge Distillation |
|---|---|---|---|
| FedGen (Zhu et al., 2021b) | ✓ | | ✓ |
| DS-FL (Itahara et al., 2021) | ✓ | | ✓ |
| FedDF (Lin et al., 2020) | ✓ | | ✓ |
| Def-KT (Li et al., 2021) | ✓[2] | | ✓ |
| QG-DSGDm-N (Lin et al., 2021) | | ✓ | |
| DSGD (Lian et al., 2017) | | ✓ | |
| $D^2$ (Tang et al., 2018) | | ✓ | |
| **QG-IDKD (Ours)** | | ✓ | ✓ |

Table 1: Prior works have proposed distillation to address non-IID data distribution in federated learning, however as far as we know the proposed work is the only approach in distributed decentralized learning leveraging distillation.

In this paper, we propose In-Distribution Knowledge Distillation (IDKD), a new technique to improve decentralized training over heterogeneous data to obtain a global solution (i.e. same model across the nodes). IDKD aims to distill heterogeneous datasets to make them more IID. Distillation (Hinton et al., 2015) is a process in which a teacher network transfers knowledge to a student network. This is often done by having the teacher network label a corpus of unlabelled public data (this is separate from the training set). The student network is trained to match the teacher labels on this corpus of data. In our problem setting, the teacher networks are other nodes in the graph and the student is the local network. By using an *unlabelled public dataset*, we distill the knowledge from each node and communicate it to its neighbors. However, with distillation, the alignment of the public and private sets is crucial for generalization performance (Hinton et al., 2015). Thus we propose the use of an Out-of-Distribution (OoD) detector during the distillation process. This improves distillation performance and model generalization while reducing communication overhead.

A conceptual overview of IDKD is visualized in Figure 1. We visualize a toy latent space of 2 neighboring nodes in a decentralized setup with non-IID data. These two nodes will learn different decision boundaries due to data skew (see Figure 1(a)). Our goal is to encourage these nodes to learn similar decision boundaries that generalize better via knowledge distillation. In the traditional knowledge distillation methods, entire public data is used regardless of its alignment with the private dataset. However, in the decentralized setting, each node generates a different set of labels for the public data due to the model/data variation. This gives rise to the issue of *label conflict* (see public samples with black dotted outlines in Figure 1(b)). To address the issues of data alignment and label conflict, we propose excluding low-confidence samples and communicating the labels of high-confidence samples (shown in red arrow in Figure 1(c)). This label exchange results in more homogeneous data distribution, which in turn results in better decision boundaries and improved generalization performance (as seen in Figure 1(d)).

To summarize our contributions,

- We propose a new distillation-based approach to handle non-IID data distribution in a decentralized setting to obtain a global solution (i.e. same model across the nodes). We focus on the dataset, rather than algorithm design.

- We show that public-private dataset misalignment reduces distillation performance. To address this we propose using an Out-of-Distribution (OoD) detector.

---

[2]In Def-KT the authors assume complete graph connectivity and thus is equivalent to a centralized setup.

- We show that the proposed IDKD framework is superior $(2 - 13\%)$ to vanilla knowledge distillation in a decentralized setting and achieves up to $4 - 8\%$ improvement over the state-of-the-art in generalization performance on heterogeneously distributed data with minimal communication overhead $(\sim 2\%)$.

## 2 Related Work

**Knowledge distillation** was proposed (Hinton et al., 2015) to reduce the computational complexity of large models. It was used to distill a larger teacher model into a smaller student model while retaining similar performance. However, distillation assumes that the training dataset $D_T$ and the public (a.k.a student) dataset $D_P$ are similar (Ahn et al., 2019; Hinton et al., 2015; Tian et al., 2020). Hence, any misalignment between the training set and the distillation set reduces generalization performance. To address this issue, in our work, we propose using an OoD detector to distill on a subset that is aligned with the local dataset improving distillation performance.

**Decentralized Learning** is a branch of distributed learning that collaboratively trains a DNN model across multiple nodes holding local data, without exchanging it. The key aspect being the peer-to-peer exchange of information in the form of model parameters or gradients (no central server). Researchers proposed Decentralized Stochastic Gradient Descent (DSGD) (Lian et al., 2017; 2018) by combining SGD with gossip algorithms (Xiao & Boyd, 2004) to enable decentralized deep learning. To improve performance researchers have since proposed versions of DSGD with local momentum (DSGDm) (Assran et al., 2019; Kong et al., 2021; Koloskova et al., 2020; Yuan et al., 2021). The above-mentioned works have demonstrated that decentralized algorithms can perform comparably to centralized algorithms on benchmark vision datasets. However, these techniques assume the data to be Independent and Identically Distributed (IID).Training DNN models in a decentralized fashion with non-IID data still remains a major challenge. There have been several efforts in the literature to address this challenge of non-IID data in a decentralized setup (Aketi et al., 2022; Esfandiari et al., 2021; Koloskova et al., 2021; Lin et al., 2021; Tang et al., 2018). Several approaches (Aketi et al., 2022; Esfandiari et al., 2021; Koloskova et al., 2021) have attempted to improve non-IID performance by manipulating the local gradients at the cost of communication overhead over DSGD. The authors of $D^2$ (Tang et al., 2018) proposed a form of bias correction technique and theoretically show that it eliminates the influence of data heterogeneity. The authors of quasi-global momentum (QG-DSGDm-N (Lin et al., 2021)) claim that their approach stabilizes training and they provide empirical evidence to show its effectiveness.

Previously discussed methods focus on decentralized optimization algorithm design. Another orthogonal direction to improve the performance with non-IID data is to explore the field of knowledge distillation (KD) to reach a better consensus across the nodes. KD has been well explored in federated learning (FL) setups with centralized server for non-IID data (Itahara et al., 2021; Jeong et al., 2018; Li & Wang, 2019; Lin et al., 2020; Zhu et al., 2021a;b). Prior works in FL (Itahara et al., 2021; Jeong et al., 2018) have leveraged KD to reduce communication. Researchers (Zhu et al., 2021b) have proposed data-free knowledge distillation to deal with non-IID data to learn a generator and avoid using a public dataset. The authors of Federated Distillation Fusion (FedDF) (Lin et al., 2020) propose a distillation framework for federated model fusion which allows clients to have heterogeneous models/data and train the server model with less communication. However, most of the KD approaches proposed for non-IID data in FL setups leverage the central server and thus, are intractable in a decentralized setup. In this paper, we explore the KD paradigm for decentralized setups. Unlike previous research, we leverage KD to approach the non-IID problem through the lens of dataset homogenization.

## 3 Method

The proposed In-Distribution Knowledge Distillation (IDKD) aims to homogenize non-IID data across the nodes to make full use of existing decentralized training schemes. IDKD decentralized training makes use of two datasets, a private training dataset $D_T^i$ local to each node $i$ and a public dataset $D_P$ (common across nodes). Using the unlabelled public dataset $D_P$, we distill the knowledge from each node and communicate it to other nodes in the graph. In our problem setting, the teacher networks are the other nodes in the

graph and the student is the local network. The proposed IDKD framework utilizes a 5-step process: (i)

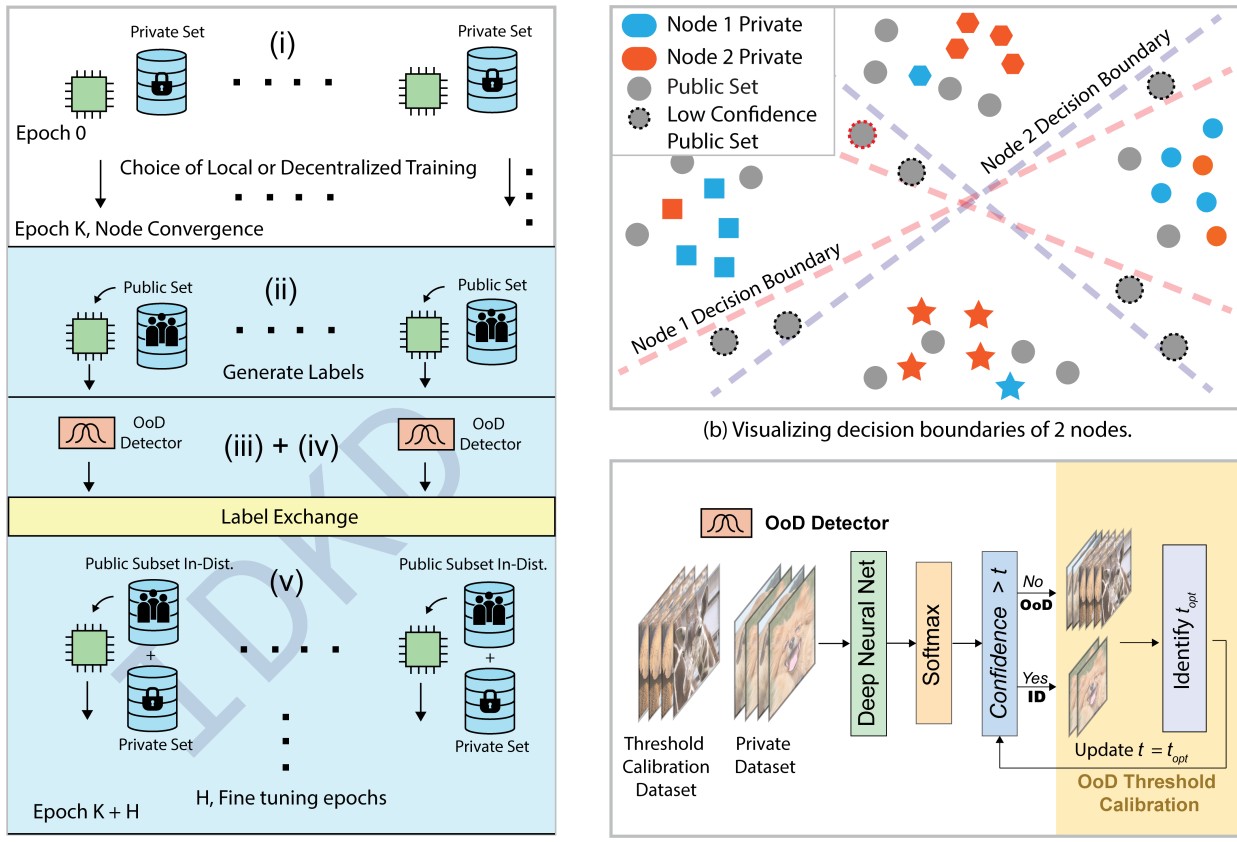

(a) Overview of the proposed IDKD framework for decen--tralized training.

(b) Visualizing decision boundaries of 2 nodes.

(c) The OoD detector used by the IDKD framework.

Figure 2: (a) Overview of the proposed IDKD framework for decentralized training. (i) Each node $i$ trains on the private dataset $D_T^i$ till convergence on a decentralized training algorithm. (ii) Next, soft labels for the public dataset are generated (iii) OoD detector is calibrated (iv) OoD detector is used to extract a subset dataset $D_{ID}^i$ that is similar to the private dataset. (v) Soft labels corresponding to $D_{ID}^i$ are exchanged between the neighbors and the models are fine-tuned on the private and public dataset subset. (b) Visualizing decision boundaries of 2 nodes. For improved KD we include ID-like data from the public set while excluding low-confidence conflicting examples. (c) The OoD detector used by the IDKD framework. A calibration dataset and the private dataset are used as OoD data and ID data respectively. This is used to identify the optimal threshold.

Initial training, (ii) Generating soft labels at each node, (iii) OoD detector calibration, (iv) In-Distribution (ID) subset generation, and (v) Label exchange and fine-tuning. The pseudocode for the IDKD framework is described by Algorithm 1 and an overview of different steps is given in Figure 2. Next, we provide details for each of these steps.

**Initial Training** Each node $i$ is trained on the private dataset $D_T^i$ till convergence using a decentralized training scheme. This corresponds to Line 3 in Algorithm 1. The IDKD framework is general and supports any user-specific decentralized training scheme. In our work, we choose QG-DSGDm-N (Lin et al., 2021) which is one of the state-of-the-art decentralized training schemes targeting non-IID data without communication overhead over DSGD.

**Local Soft Label Generation** After the initial training process, we forward propagate the public dataset $D_P$ on each node to gather the corresponding soft labels (shown in line 5 in Algorithm 1). A soft label $s_p$ is a vector that has a probability/likelihood score for each output class.

**OoD Detector** In vanilla distillation, the labels generated by the teacher model on the public dataset are used to train the student model. The key to effective distillation is the alignment of the public and private datasets. Further, specific to decentralized KD, we observe that the low-confidence public data samples (seen in Figure 2(b) as dotted samples) usually fall on different sides of the decision boundary for different nodes i.e., have conflicting labels. This property can be attributed to the differences in the decision boundary across the nodes arising from data skew. For example in Figure 2(b), the sample with the red dotted outline is Class Hexagon according to Node 1 while it is Class Square according to Node 2. This creates label conflicts when communicating the labels to the neighbors. Specifically, two cases can occur - (a) both nodes have low confidence in a sample, in which case the sample should be ignored (b) one node is confident while the other is not. In this case, the high confidence label is retained which is potentially labeled from a model with more samples (for that class). Hence, to address both issues (dataset alignment and labeling conflict), we use an Out-of-Distribution (OoD) detector to identify a subset of $D_P$ that aligns with the private training set $D_T^i$ and excludes low-confidence samples.

The *OoD detector* identifies the samples in the public set $D_P$ that look like In-Distribution (ID) data to the local model on each node. Among different OoD detectors proposed in literature (Hendrycks & Gimpel, 2017; Liu et al., 2020b; Ravikumar & Roy, 2022; Ravikumar et al., 2023; Yang et al., 2023), we choose the maximum softmax probability (MSP) detector (Hendrycks & Gimpel, 2017) because of its simplicity and low computational overhead. Figure 2(c) visualizes the MSP detector. The samples with confidence greater than a threshold $t$ are classified as ID by the MSP detector

The hyperparameter $t$ (threshold) is tuned based on the True Positive Rate (TPR) of the OoD detector. The ideal threshold $t_{opt}$ is the point at which the TPR is maximized while minimizing the false positive rate (FPR) (Fawcett, 2006). To calculate TPR and FPR, we use a calibration dataset $D_C$ (we use the public dataset but another dataset may also be used) as OoD data and the private dataset $D_T^i$ as ID data. Subsequently, the threshold $t$ was adjusted iteratively to maximize TPR while minimizing FPR to identify the optimal value $t_{opt}$. This corresponds to Line 6 in Algorithm 1 and is visualized in Figure 2(c).

---

**Algorithm 1** IDKD framework at each node

---

**Input:** Public dataset $D_P$, local training dataset $D_T^i$, local validation dataset $D_V^i$, number of nodes $N$, OoD calibration dataset $D_C$, Decentralized training algorithm $\mathcal{A}$, OoD Detector $OoD$
**Output:** IDKD trained model $\mathcal{M}$
1:   $D_{Tr}^i \leftarrow D_T^i$
2:   **for** $e \in \{1, 2, \cdots \text{total epochs}\}$ **do**
3:     $\mathcal{M} \leftarrow Train(\mathcal{A}, D_{Tr}^i)$
4:     **if** $e >$ local convergence & $e$ mod $k = 0$ **then**
5:       $KD_P \leftarrow GenerateSoftLabels(D_P)$
6:       $t_{opt} \leftarrow Optimal(D_C, D_V^i, OoD)$
7:       $D_{ID}^i = \{(p, s_p) \in KD_p : \max(s_p) > t_{opt}\}$
8:       $D_{ID}^N = \varnothing$
9:       **for** $j \in Neighbors(i)$ **do**
10:         # Only labels are sent
11:         $D_{ID}^j \leftarrow SendReceive(D_{ID}^i, j)$
12:         $D_{ID}^N \leftarrow D_{ID}^N \cup \{D_{ID}^j\}$
13:       **end for**
14:       $D_{ID} \leftarrow LabelAverage(D_{ID}^N)$
15:       $D_{Tr}^i \leftarrow D_{ID} \cup D_T^i$
16:     **end**
17: **end for**
18: **return** $\mathcal{M}$

---

**In-Distribution (ID) Subset Generation** In this step, we use the calibrated OoD detector on each node to identify a subset of the public dataset $D_P$ that aligns with the private training set $D_T^i$. On each node, the generated soft labels from the public dataset are processed by the MSP OoD detector. The samples from the public data that are classified as In-Distribution (ID) by the OoD detector are added to the local

subset $D_{ID}^i$. The samples from the public dataset with maximum softmax probability greater than $t_{opt}$ are classified as ID samples. The pseudocode for ID subset generation is shown in line 7 in Algorithm 1.

**Label Exchange and Fine Tuning** Once the In-Distribution subset ($D_{ID}^i \subset D_P$) is identified for each node, the corresponding soft labels are communicated to the neighbors of each node. Each node now has access to the soft labels of the In-Distribution subsets corresponding to its neighbors in the graph (Lines 8 - 13). The soft labels are averaged to obtain the final training labels on the distilled ID datasets (Line 14). This process of label exchange with the neighbors is repeated every $k$ epochs and hence the information flows across the graph. Every exchange the training set is updated $D_{Tr}^i \leftarrow D_{ID} \cup D_T^i$ (refer line 15 in Algorithm 1). Where $D_T^i$ is the original private set at the beginning of training. This process provides desired data homogenization at each local node.

## 4 Experiments

### 4.1 Setup

In a decentralized training setup, each node has a local private dataset. This dataset is not shared between the nodes. The nodes are connected in a fixed or time-varying graph topology (such as a ring, chain, etc.). The aim of such a setup is to converge to a global model while only being able to communicate with immediate neighbors. Prior research (Dandi et al., 2022) has shown that the specifics of such a network (i.e. network configuration) play an important role in convergence. In our work, we analyze our method on 2 different graph topologies – Ring and Social Network. The ring topology is an important consideration because it is a topology where information traversal between the nodes is among the slowest. Social graphs are of interest because it lets us consider networks with a lower spectral gap (i.e. better connectivity) than a ring network. Thus, making our analysis more complete. For the social network, we use the Florentine families graph (Breiger & Pattison, 1986). To show the scalability of the proposed method, we present the experiments with varying graph sizes (8 to 32 nodes). Further, to show the effect of data heterogeneity, we run experiments with the Dirichlet parameter $\alpha$ set to $1, 0.1, 0.05$. The $\alpha$ value of 0.1 is considered significantly heterogeneous.

**Datasets and models:** For our experiments, we use ResNet (He et al., 2016) architecture on CIFAR-10, CIFAR-100 (Krizhevsky et al., 2009), and Imagenette (Howard, 2018) datasets. We choose TinyImageNet (Li et al., 2015), LSUN (Yu et al., 2015), and Uniform-Noise as the distillation datasets (public dataset). We use ResNet20 with EvoNorm (Liu et al., 2020a), i.e. the Batch Norm layers of ResNet20 are replaced with EvoNorm as Batch Norm (Ioffe & Szegedy, 2015) is shown to fail in a decentralized setting with heterogeneous data distribution (Andreux et al., 2020; Hsieh et al., 2020). We report the test accuracy of the averaged/consensus model. All our experiments are conducted for three random seeds and the mean and standard deviation results are reported. For CIFAR datasets, we train the models for 300 epochs with a mini-batch size of 32 per node. When using the IDKD framework, the label exchange is done at epoch 240. The learning rate is decayed by 10 when the number of training epochs reaches 60% and 80% of the total number of epochs.

**Non-IID data distribution:** We sample from the Dirichlet distribution to partition the data (non-overlapping) across nodes in a non-IID fashion. The Dirichlet distribution is a multivariate generalization of the beta distribution. Once the dataset is partitioned, the data is never shuffled across the nodes. The IIDness of the data partition is controlled by the Dirichlet parameter $\alpha$. Larger the value of $\alpha$, the more IID the data partition and vice versa. When $\alpha$ is small, it is more likely that each node has data samples from only one class. The details of implementation, compute resources used and hyperparameters are provided along with the source code in the supplementary material.

### 4.2 Results

We evaluate the performance of the proposed IDKD methodology on various datasets and graph topologies and compare it against the current state-of-the-art decentralized learning algorithm. We chose QG-DSGDm-N (Lin et al., 2021) as our primary baseline as it achieves state-of-the-art performance on heterogeneous data

| Dataset | Method | Ring ($n = 16$) | | | Ring ($n = 32$) | | |
|---|---|---|---|---|---|---|---|
| | | $\alpha = 1$ | $\alpha = 0.1$ | $\alpha = 0.05$ | $\alpha = 1$ | $\alpha = 0.1$ | $\alpha = 0.05$ |
| CIFAR-10 | SGD-Centralized (Goyal et al., 2017) | | $90.94 \pm 0.03$ | | | $90.85 \pm 0.17$ | |
| | DSGD (Lian et al., 2017) | $86.61 \pm 0.22$ | $75.40 \pm 3.28$ | $51.96 \pm 5.36$ | $83.29 \pm 0.66$ | $59.42 \pm 7.13$ | $47.67 \pm 7.97$ |
| | Relay-SGD (Vogels et al., 2021) | $88.45 \pm 0.15$ | $79.11 \pm 1.17$ | $68.15 \pm 2.01$ | $85.74 \pm 0.48$ | $76.39 \pm 2.68$ | $67.11 \pm 7.54$ |
| | QG-DSGDm-N (Lin et al., 2021) | $88.98 \pm 0.11$ | $82.94 \pm 2.10$ | $73.41 \pm 4.12$ | $\mathbf{89.31 \pm 0.11}$ | $82.23 \pm 1.40$ | $72.41 \pm 3.45$ |
| | QG-DSGDm-N + KD (Li et al., 2021) | $88.85 \pm 0.52$ | $85.34 \pm 0.77$ | $77.18 \pm 2.00$ | $87.86 \pm 0.42$ | $83.16 \pm 0.28$ | $77.66 \pm 2.53$ |
| | **QG-IDKD (Ours)** | $\mathbf{89.53 \pm 0.29}$ | $\mathbf{86.47 \pm 0.43}$ | $\mathbf{81.70 \pm 1.74}$ | $88.67 \pm 0.19$ | $\mathbf{85.55 \pm 0.23}$ | $\mathbf{79.43 \pm 0.72}$ |
| CIFAR-100 | SGD-Centralized (Goyal et al., 2017) | | $65.96 \pm 0.15$ | | | $65.62 \pm 0.51$ | |
| | DSGD (Lian et al., 2017) | $59.17 \pm 0.25$ | $54.00 \pm 0.46$ | $48.20 \pm 1.07$ | $50.61 \pm 0.81$ | $49.20 \pm 0.74$ | $46.87 \pm 0.70$ |
| | Relay-SGD (Vogels et al., 2021) | $61.77 \pm 1.01$ | $54.11 \pm 0.68$ | $50.15 \pm 0.76$ | $57.38 \pm 0.79$ | $52.75 \pm 1.35$ | $47.73 \pm 1.14$ |
| | QG-DSGDm-N (Lin et al., 2021) | $\mathbf{63.11 \pm 0.13}$ | $53.36 \pm 1.84$ | $47.09 \pm 2.24$ | $\mathbf{61.75 \pm 0.23}$ | $56.96 \pm 0.54$ | $50.00 \pm 2.06$ |
| | QG-DSGDm-N + KD (Li et al., 2021) | $41.89 \pm 0.23$ | $48.53 \pm 1.18$ | $42.69 \pm 1.54$ | $48.46 \pm 1.13$ | $47.10 \pm 2.00$ | $41.21 \pm 2.11$ |
| | **QG-IDKD (Ours)** | $62.12 \pm 0.93$ | $\mathbf{56.65 \pm 0.80}$ | $\mathbf{52.92 \pm 1.06}$ | $61.04 \pm 0.33$ | $\mathbf{57.44 \pm 0.69}$ | $\mathbf{54.16 \pm 1.49}$ |

Table 2: Evaluating the performance of the proposed framework against existing decentralized training schemes on CIFAR-10 and CIFAR-100 datasets using ResNet20-EvoNorm on two different network sizes. Please note that for the SGD-Centralized, we report the results for a random IID data distribution. For QG-IDKD (ours) TinyImageNet was used as the public dataset.

without incurring communication overhead over DSGD (Lian et al., 2017). For an exhaustive analysis, we present other baselines that incur no communication overhead such as (QG-DSGDm-N) with vanilla KD (Li et al., 2021) and Relay-SGD (Vogels et al., 2021). Further. we include SGD-Centralized (Goyal et al., 2017) as a reference to show the upper bound of performance. Note that for the SGD-Centralized, that data is randomly distributed across the nodes (IID data). All the methods are trained on identical initial seeds, hyper-parameters, and data distributions. Note that the key hyper-parameter in KD methods is the distillation temperature and it needs to be tuned appropriately. We vary the distillation temperature from 1 - 1000 and report the best-performing distillation model (temperature 10). The performance results are presented in Table 2. From Table 2, we observe that the proposed IDKD (TinyImageNet used as public dataset) performs significantly better than existing schemes, especially as skew increases (i.e. $\alpha$ decreases). On CIFAR datasets with a skew of 0.05, IDKD improves the accuracy by $4 - 8\%$ compared to QG-DSGDm-N and $2 - 13\%$ compared to vanilla KD. We draw two main conclusions from Table 2. First, the proposed IDKD method performs the best and outperforms vanilla knowledge distillation. Second, the use of the OoD detector is the key to the observed performance boost.

**Graph Topologies:** In this section, we explore two different graph topologies for the decentralized setup i.e., ring and social network. To add variety to the datasets used, we use the ImageNette dataset (Howard, 2018). Similar to the previous section we train a ResNet20-EvoNorm architecture and use TinyImageNet as the public dataset. The results presented in Table 3 show that the trends for the ImageNette dataset are the same as the CIFAR datasets. The proposed technique performs better than QG-DSGDm-N. The exception being Ring of 8 with $\alpha = 0.1$. Thus, the takeaway from this experiment is that IDKD performs better on various graph topologies.

**Node Failures (Time-Varying Graphs):** In earlier discussions, we focused on static graphs. To demonstrate the real-world relevance of this method, we examine node failures, which lead to time-varying graphs. Previous baselines lack convergence guarantees under time-varying graphs. Therefore, this section presents findings using DSGD and SGP (Assran et al., 2019), both of which are known to converge under time-varying graphs. We implement IDKD with DGSD to highlight improvements achieved through data homogenization. For simulating node failures, we introduced random communication failures at rates of $1\%$ and $2\%$ (a failure is triggered if a uniformly generated random number between $0 - 1$ falls below the specified failure rate). The results shown in Table 4 validate that IDKD enhances DSGD's effectiveness via data homogenization

| Dataset | Method | Ring ($n = 8$) | | Social Network ($n = 15$) | |
|---|---|---|---|---|---|
| | | $\alpha = 0.1$ | $\alpha = 0.05$ | $\alpha = 0.1$ | $\alpha = 0.05$ |
| CIFAR10 | DSGD | $76.12 \pm 5.29$ | $62.99 \pm 3.14$ | $76.81 \pm 1.68$ | $69.42 \pm 4.29$ |
| | Relay-SGD | $77.18 \pm 1.92$ | $62.67 \pm 5.95$ | $75.99 \pm 1.82$ | $69.95 \pm 1.19$ |
| | QG-DSGDm-N | $77.79 \pm 1.19$ | $65.54 \pm 3.37$ | $77.79 \pm 1.19$ | $68.29 \pm 6.71$ |
| | QG-DSGDm-N + KD | $81.23 \pm 0.14$ | $76.87 \pm 1.05$ | $84.08 \pm 0.22$ | $75.03 \pm 0.18$ |
| | **QG-IDKD (Ours)** | $\mathbf{84.58 \pm 1.40}$ | $\mathbf{77.83 \pm 1.98}$ | $\mathbf{84.72 \pm 0.75}$ | $\mathbf{80.33 \pm 2.50}$ |
| CIFAR100 | DSGD | $48.67 \pm 2.65$ | $43.15 \pm 3.47$ | $55.11 \pm 1.91$ | $50.41 \pm 2.36$ |
| | Relay-SGD | $52.44 \pm 0.98$ | $44.38 \pm 2.75$ | $54.75 \pm 0.10$ | $50.00 \pm 0.77$ |
| | QG-DSGDm-N | $49.06 \pm 1.30$ | $44.80 \pm 1.39$ | $52.14 \pm 2.54$ | $46.31 \pm 2.21$ |
| | QG-DSGDm-N + KD | $52.21 \pm 0.54$ | $49.67 \pm 0.36$ | $51.60 \pm 2.09$ | $48.97 \pm 0.83$ |
| | **QG-IDKD (Ours)** | $\mathbf{55.01 \pm 0.78}$ | $\mathbf{50.38 \pm 1.53}$ | $\mathbf{56.02 \pm 1.81}$ | $\mathbf{51.51 \pm 2.45}$ |
| ImageNette | DSGD | $48.60 \pm 8.89$ | $48.09 \pm 11.45$ | $42.78 \pm 5.51$ | $48.51 \pm 5.57$ |
| | Relay-SGD | $72.88 \pm 4.29$ | $70.95 \pm 5.22$ | $78.08 \pm 2.79$ | $76.80 \pm 3.79$ |
| | QG-DSGDm-N | $\mathbf{78.30 \pm 1.33}$ | $73.31 \pm 5.39$ | $77.09 \pm 2.37$ | $75.39 \pm 2.80$ |
| | QG-DSGDm-N + KD | $49.67 \pm 0.36$ | $52.21 \pm 0.54$ | $51.60 \pm 2.09$ | $67.74 \pm 1.13$ |
| | **QG-IDKD (Ours)** | $74.18 \pm 3.01$ | $\mathbf{74.60 \pm 0.45}$ | $\mathbf{78.24 \pm 4.49}$ | $\mathbf{76.51 \pm 2.96}$ |

Table 3: Accuracy of ResNet20-EvoNorm trained on different graph topologies. For QG-IDKD (ours) and QG-DSGDm-N + KD, TinyImageNet was used as the public dataset.

| Failure Rate | Method | Dataset | | |
|---|---|---|---|---|
| | | CIFAR-10 | CIFAR-100 | ImageNette |
| 1% | DSGD | $67.61 \pm 3.82$ | $42.21 \pm 3.24$ | $55.05 \pm 2.10$ |
| | SGP | $57.52 \pm 0.97$ | $34.04 \pm 5.52$ | $\mathbf{64.91 \pm 5.41}$ |
| | **DSGD-IDKD (Ours)** | $\mathbf{78.78 \pm 1.86}$ | $\mathbf{43.46 \pm 2.25}$ | $62.30 \pm 3.29$ |
| 2% | DSGD | $64.18 \pm 6.44$ | $44.40 \pm 1.88$ | $64.23 \pm 4.85$ |
| | SGP | $60.18 \pm 2.09$ | $34.38 \pm 4.77$ | $61.02 \pm 6.65$ |
| | **DSGD-IDKD (Ours)** | $\mathbf{79.96 \pm 0.28}$ | $\mathbf{45.06 \pm 2.47}$ | $\mathbf{64.49 \pm 1.26}$ |

Table 4: Accuracy of ResNet20-EvoNorm trained on a ring (8 nodes, $\alpha = 0.05$) topology with CIFAR-10, CIFAR-100 and ImageNette datasets with 1% and 2% communication failure.

even under node failures. Interstingly, we observed that at the specified communication failure rates, such disruptions act as a form of regularization, boosting the performance of DSGD (see Table 3 vs. Table 4).

**Impact of Public Dataset Selection:** The performance improvements achieved by knowledge distillation methods are highly dependent on the selection of a suitable public dataset. To reduce this dependence, the proposed IDKD framework utilizes an OoD detector. To evaluate the impact of public datasets, we run the IDKD framework with various public datasets such as LSUN (Yu et al., 2015) and Uniform Noise. Table 5 compares the vanilla KD with the proposed IDKD framework on these public datasets. Note that we use a subset of the LSUN dataset with 300,000 samples for distillation. The performance trends of the proposed method are the same as observed when using TinyImageNet as the distillation dataset. Thus, IDKD is able to choose a subset of the distillation dataset that aligns with the private dataset improving the KD performance. Another aspect is the amount of public data used, we found that more data generally helps upto a limit (for details of this experiment see the Appendix).

**Visualizing effect of IDKD:** To visualize the effect of IDKD on data heterogeneity, we compare the class distribution with and without IDKD. In particular, we compute the normalized number of samples per class on a single node with and without IDKD. To remove other influences, we collect the class distributions from the same run. That is, for the without IDKD sample distribution (pre-IDKD), we save the initial distribution

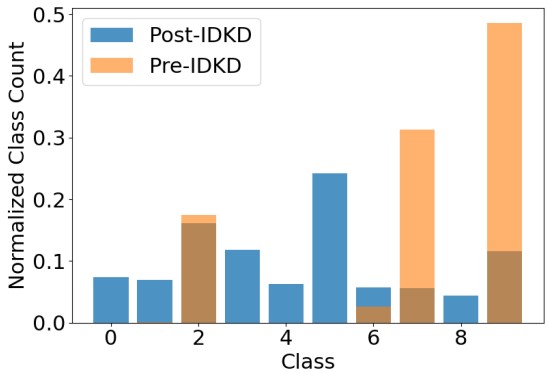

(a) Visualizing normalized class distribution pre-IDKD and post-IDKD from a single node of a 16-nodes ring graph. CIFAR-10 with $\alpha = 0.1$ distilled with TinyImageNet.

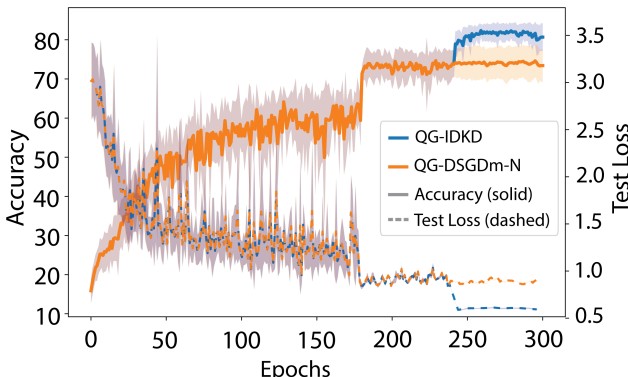

(b) Test accuracy, test loss (mean and std) vs epochs of the proposed IDKD framework and QG-DSGDm-N for training CIFAR-10 dataset on ResNet20-EvoNorm, Ring 16 with a skew of 0.05.

Figure 3: Visualizing the results of the proposed IDKD method (a) on the data distribution pre and post IDKD (b) comparing the convergence of IDKD vs DSGDm-N.

sampled from the Dirichlet distribution. For the distribution with IDKD (post-IDKD), we collect the soft labels along with the initial distribution. Since we use soft labels for the public ID dataset $D_{ID}$, the soft labels are counted for all the classes where the value is non-zero. For example, if a soft label is $[0.1, 0.2, 0.7, 0, 0]$, we add 0.1 to the sample count for class 1, 0.2 to the sample count for class 2, and so on. This is repeated for all the samples and the count is rounded to the nearest whole number. This count is then normalized. Figure 3a visualizes the normalized samples per class with and without IDKD while training CIFAR-10 on 16-ring with a skew of 0.1. We clearly see a reduction in data skew (more IID) after performing IDKD. Pre-IDKD we can see that there are no samples for classes 0, 1, 3, 4, 5, and 8. We observe that the samples are more evenly distributed post-IDKD.

**Convergence:** To evaluate the convergence performance of QG-IDKD, we plot test accuracy and test loss error vs epochs (Figure 3b). Figure 3b plots the average accuracy and standard deviation for all nodes in the graph. Since we use the same seeds and random states QG-IDKD and QG-DSGDm-N perform similarly until the homogenization step at epoch 240, after which QG-IDKD outperforms QG-DSGDm-N. The results were plotted for ResNet20-EvoNorm, ring 16 with a skew of 0.05 for the CIFAR-10 dataset. Figure 3b also plots the results for the test loss for the same configuration and we see that similar to test accuracy there is a significant reduction in test loss after the homogenization step by IDKD.

| Dataset | Method | Accuracy on Distillation Dataset | | |
|---------|--------|-------------|------|---------------|
| | | TinyImageNet | LSUN | Uniform-Noise |
| CIFAR-10 | QG-DSGDm-N + KD | $77.18 \pm 2.00$ | $75.48 \pm 1.90$ | $74.15 \pm 3.05$ |
| | **QG-IDKD (Ours)** | $81.70 \pm 1.74$ | $78.56 \pm 2.02$ | $77.45 \pm 3.77$ |
| CIFAR-100 | QG-DSGDm-N + KD | $42.69 \pm 1.54$ | $46.43 \pm 3.22$ | $45.80 \pm 1.44$ |
| | **QG-IDKD (Ours)** | $52.92 \pm 1.06$ | $51.62 \pm 1.15$ | $48.78 \pm 2.02$ |
| ImageNette | QG-DSGDm-N + KD | $62.29 \pm 7.66$ | $57.55 \pm 6.94$ | $60.87 \pm 3.66$ |
| | **QG-IDKD (Ours)** | $74.60 \pm 0.45$ | $73.63 \pm 0.86$ | $75.96 \pm 0.97$ |

Table 5: ResNet20-EvoNorm distilled with a subset of the LSUN, TinyImageNet, and Uniform-Noise. We use the Ring topology with 16 nodes for CIFAR-10 and CIFAR-100 and 8 nodes for ImageNette. Dirichlet parameter $\alpha = 0.05$.

| Dataset | Method | Ring ($n = 16$) | | | Ring ($n = 32$) | | |
|---|---|---|---|---|---|---|---|
| | | $\alpha = 1$ | $\alpha = 0.1$ | $\alpha = 0.05$ | $\alpha = 1$ | $\alpha = 0.1$ | $\alpha = 0.05$ |
| CIFAR-10 | SGD-Centralized | | $90.94 \pm 0.03$ | | | $90.85 \pm 0.17$ | |
| | QG-DSGDm-N | $88.98 \pm 0.11$ | $82.94 \pm 2.10$ | $73.41 \pm 4.12$ | $\mathbf{89.31 \pm 0.11}$ | $82.23 \pm 1.40$ | $72.41 \pm 3.45$ |
| | QG-DSGDm-N + KD | $88.85 \pm 0.52$ | $85.34 \pm 0.77$ | $77.18 \pm 2.00$ | $87.86 \pm 0.42$ | $83.16 \pm 0.28$ | $77.66 \pm 2.53$ |
| | **QG-IDKD (Ours)** | $\mathbf{89.02 \pm 0.28}$ | $\mathbf{86.84 \pm 0.57}$ | $\mathbf{81.69 \pm 2.19}$ | $88.89 \pm 0.39$ | $\mathbf{84.96 \pm 0.24}$ | $\mathbf{79.81 \pm 0.95}$ |
| CIFAR-100 | SGD-Centralized | | $65.96 \pm 0.15$ | | | $65.62 \pm 0.51$ | |
| | QG-DSGDm-N | $\mathbf{63.11 \pm 0.13}$ | $53.36 \pm 1.84$ | $47.09 \pm 2.24$ | $\mathbf{61.75 \pm 0.23}$ | $56.96 \pm 0.54$ | $50.00 \pm 2.06$ |
| | QG-DSGDm-N + KD | $41.89 \pm 0.23$ | $48.53 \pm 1.18$ | $42.69 \pm 1.54$ | $48.46 \pm 1.13$ | $47.10 \pm 2.00$ | $41.21 \pm 2.11$ |
| | **QG-IDKD (Ours)** | $61.84 \pm 1.12$ | $\mathbf{56.56 \pm 0.82}$ | $\mathbf{52.83 \pm 0.59}$ | $60.53 \pm 0.46$ | $\mathbf{57.02 \pm 0.74}$ | $\mathbf{53.97 \pm 0.82}$ |

Table 6: Performance of the proposed framework against existing decentralized training schemes on CIFAR-10 and CIFAR-100 datasets using ResNet20-EvoNorm on two different network sizes under iso-iteration setting. Thus, the compute overhead is 0. Please note that for the SGD-Centralized, we report the results for a random IID data distribution. For QG-IDKD (ours) TinyImageNet was used as the public dataset. The results are almost identical to the ones presented in Table 2 which are under iso-epoch setting.

**Computation Overhead:** Here we analyze the compute cost of the proposed method. For compute analysis we consider iso-iteration performance results. Before we analyze the results, we briefly discuss iso-iteration. Each iteration of the algorithm uses a minibatch of data during backpropagation to compute the gradients. Thus under iso-iterations (i.e. the same number of iterations) and fixed mini-batch size (was set to 32 for these experiments), the algorithms under consideration have the same compute cost. Thus, for the results presented in Table 6 the compute overhead of our method between 1.5 % (best case) - 15% (worst case), average case $\approx 4.5\%$ over the baseline algorithm (depends on how many samples from the public dataset were used during training; see the Appendix for details), and the communication overhead remains the same as discussed before. Further, it is interesting to note that the results of iso-iteration IDKD (Table 6) show similar performance uplift as iso-epoch (Table 2). Further, to add a more varied dataset result we also trained FashionMNIST (a dataset dissimilar to CIFAR10/100 or ImageNette) with iso-iterations until QG-DSGDm-N convergence. For this this experiment the mini-batch size was set to 32, on 16 node-ring with $\alpha = 0.05$, for $\approx 35400$ iterations. QG-DSGDm-N obtained $77.04 \pm 0.02$ while QG-IDKD achieved $80.71 \pm 0.06$ when using TinyImagenet as public dataset. Clearly, from the FashionMNIST results and Table 6 we see that IDKD improves classification performance while having minimal compute overhead.

**Communication Cost:** It is common to report bytes transmitted per iteration (Koloskova et al., 2020) to measure communication cost. Note that a training epoch contains multiple iterations. We report the communication cost per iteration for CIFAR-10 and CIFAR-100 datasets at iso-iteration. We compare QG-DSGDm-N and the proposed IDKD framework. We use a ResNet20-EvoNorm for these experiments and report MiB (MebiBytes, 1 MiB = $1024^2$ bytes) per iteration in Table 7. Table 7 reports the mean and standard deviation from three different seeds. The variation in the number of MiB per iteration for the proposed technique is due to the fact that different seeds/runs have different data distribution, thus the number of labels that are transmitted between nodes varies. From Table 7 we see that the proposed method has very minimal overhead in terms of communication cost ($\sim 2\%$ on average). This is because the communication cost of label exchange (very small) is amortized across all iterations. Further, in terms of cumulative communication cost, IDKD will add more communication due to the increased number of iterations arising from the larger training set. However, with QG-IDKD we need fewer epochs which compensates for more data. This is seen in Figure 3b, where maximum performance is achieved within 10 epochs (i.e. by epoch 250) of data homogenization.

**Ablation Study:** An ablation study was performed to assess the effects of the components of the proposed IDKD method, utilizing the baseline algorithm QG-DSGDm-N, QG-DSGDm-N with KD (knowledge distillation), and QG-IDKD (which incorporates KD and an OoD detector). The results are shown in Figure 4 which compares the performance between the baseline QG-DSGDm-N, QG-DSGDm-N with KD, and QG-IDKD. Furthermore, we examined the impact of initializing with a pretrained model using the public dataset-TinyImageNet, denoted as 'Pretrained + QG-DSGDm-N' in blue. The experiments were conducted

| Dataset | Method | MiB per iter | |
| --- | --- | --- | --- |
| | | Ring (n = 16) | Ring (n = 32) |
| CF-10 | QG-DSGDm-N | 3.13 | 3.13 |
| | QG-IDKD (Ours) | $3.33 \pm 0.05$ | $3.18 \pm 0.09$ |
| CF-100 | QG-DSGDm-N | 3.19 | 3.19 |
| | QG-IDKD (Ours) | $3.20 \pm 0.01$ | $3.21 \pm 0.02$ |

Table 7: Comparing communication cost in MiB per iteration for the proposed method and QG-DSGDm-N ($\alpha = 0.1$).

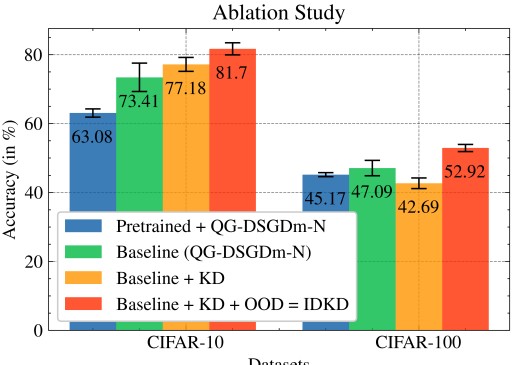

Figure 4: Ablation study results on CIFAR-10 and CIFAR-100 when using $\alpha = 0.05$ and a 16 node ring network topology.

| Graph | Dataset | Method | $\alpha = 0.1$ | $\alpha = 0.05$ |
| --- | --- | --- | --- | --- |
| Ring (n = 64) | CIFAR-10 | QG-DSGDm-N | $70.65 \pm 1.77$ | $66.37 \pm 2.89$ |
| | | **QG-IDKD (Ours)** | $\mathbf{79.74 \pm 0.25}$ | $\mathbf{76.66 \pm 1.21}$ |
| | CIFAR-100 | QG-DSGDm-N | $53.26 \pm 0.55$ | $48.76 \pm 1.09$ |
| | | **QG-IDKD (Ours)** | $\mathbf{56.91 \pm 0.60}$ | $\mathbf{54.07 \pm 0.38}$ |

Table 8: Performance of IDKD against QG-DSGDm-N on CIFAR-10 and CIFAR-100 datasets using ResNet20-EvoNorm under iso-iterations. TinyImageNet was used as the public dataset.

with $\alpha = 0.05$ on a 16-node ring topology. From Figure 4, we see that solely using KD does not consistently enhance performance compared to the baseline, but incorporating an OoD detector consistently does. Interestingly, pre-training appears to reduce performance. Additional results can also be found in Table 2.

**Scalability:** Here we analyze the scalability of IDKD as the number of nodes increases. In addition to 8, 16, and 32-node results in Tables 2 and 6, we provide 64-node iso-iteration results in Table 8. From Table 6 and Table 8, we see that as the number of nodes increases, all methods lose performance on ring networks (also observed in other related work on generic graphs, this is due to decreased speed of information travel as graph size increases). However, we observe that IDKD consistently outperforms other techniques in all cases, thus IDKD scales well as the number of nodes increase.

**IDKD and Federated Learning:** This paper introduces the IDKD framework, which is designed for non-IID decentralized learning. However, it may also be suitable for federated learning. It should be noted that federated systems are a subset of decentralized systems. For instance, a decentralized system configured in a star topology without data at the central node mirrors a federated learning framework. Results from applying IDKD to model sharing approaches are presented in Tables 2 and 4. In addition to applying IDKD on QG-DSGDm-N, Table 4 shows the results of applying IDKD on DSGD. This shows that IDKD can be readily adapted to federated learning scenarios aimed at minimizing communication rounds.

## 5 Conclusion

Decentralized learning methods achieve state-of-the-art performance on various vision benchmarks when the data is distributed in an IID fashion. However, they fail to achieve the same when the data distribution among the nodes is heterogeneous. In this paper, we propose In-Distribution Knowledge Distillation (IDKD), a novel approach to deal with non-IID data through knowledge distillation. The proposed method aims to

homogenize the data distribution on each node through the aggregation of the distilled labels on a subset of the public dataset. In particular, each node identifies and labels a subset of the public data that is a proxy to its local (private) dataset with the help of its local model and an OoD detector. The distilled versions of the local datasets are aggregated in a peer-to-peer fashion and the local models are then fine-tuned on the combined corpus of the distilled datasets along with the local dataset. We show that the proposed IDKD framework is superior to vanilla knowledge distillation (2 - 13%) in a decentralized setting. Our experiments on various datasets and graph topologies show that the proposed IDKD framework achieves up to 8% improvement in performance over the state-of-the-art techniques on non-IID data with minimal communication overhead ($\sim 2\%$).

## Acknowledgments

This work was supported in part by, the Center for the Co-Design of Cognitive Systems (CoCoSys), a DARPA-sponsored JUMP 2.0 center, the Semiconductor Research Corporation (SRC), and the National Science Foundation.

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

## Appendix

Here we present additional details about the experiments. To improve reproducibility, the source code is provided in the *IDKD_Source_Code* directory of the accompanying zip file.

### Dataset Statistics

To evaluate the performance of the proposed IDKD framework, we used various image classification benchmarks. To improve reproducibility we provide more details on the datasets and split used for our experimental evaluation. For all the experiments, we used a 10% train-validation split to identify the hyperparameters. Once the hyperparameters were tuned on the validation set, we re-ran the experiment on the complete training set with the identified hyperparameters. Details of the dataset sizes are reported in Table 9. Please note for the LSUN (Yu et al., 2015) dataset we used the first 300,000 samples subset for our experiments. This was done because LSUN is a very large dataset and would require significant compute resources to use the complete training dataset. For training the neural nets the image size for CIFAR-10 and CIFAR-100 were $3 \times 32 \times 32$ (C × H × W) and for ImageNette was $3 \times 64 \times 64$. During distillation, the distilling dataset was cropped to the same size as the training dataset.

| Dataset | Train Set Size | Validation Set Size | Test Set Size |
|---------|----------------|---------------------|---------------|
| CIFAR-10 | 45,000 (90%) | 5,000 (10%) | 10,000 |
| CIFAR-100 | 45,000 (90%) | 5,000 (10%) | 10,000 |
| ImageNette | 8,522 (90%) | 947 (10%) | 3925 |
| TinyImageNet* | 100,000 (100%) | - | - |
| LSUN* | 300,000 (100%) | - | - |
| Uniform Noise | 150,000 (100%) | - | - |

Table 9: Training, Validation and Test set sizes for the datasets used. Datasets with * were used as public dataset for distillation hence all the images from the training set were used without partitioning it to a validation set.

| Dataset | Method | LR | $\beta$ | BS | $N$ | Topology | Weight Decay | $\gamma$ |
|---------|--------|----|---------|----|----|----------|--------------|----------|
| | QG-DSGDm-N / QG-IDKD | 0.5 | 0.9 | 32 | 16, 32 | Ring | 1.00E-04 | N/A |
| CIFAR-10 | DSGD | 0.1 | 0.9 | 32 | 16, 32 | Ring | 5.00E-04 | 1 |
| | Relay-SGD | 0.1 | 0.9 | 32 | 16, 32 | Chain | 5.00E-04 | 1 |
| | QG-DSGDm-N / QG-IDKD | 0.5 | 0.9 | 32 | 16, 32 | Ring | 1.00E-04 | N/A |
| CIFAR-100 | DSGD | 0.1 | 0.9 | 32 | 16, 32 | Ring | 5.00E-04 | 1 |
| | Relay-SGD | 0.1 | 0.9 | 32 | 16, 32 | Chain | 5.00E-04 | 1 |
| | QG-DSGDm-N / QG-IDKD | 0.5 | 0.9 | 32 | 8, 15 | Ring, Social | 1.00E-04 | N/A |
| ImageNette | DSGD | 0.1 | 0.9 | 32 | 8, 15 | Ring, Social | 5.00E-04 | 1 |
| | Relay-SGD | 0.1 | 0.9 | 32 | 16, 32 | Chain | 5.00E-04 | 1 |

Table 10: Hyper parameters used for training the models on various datasets and decentralized algorithms. Dirichlet parameter $\alpha$, Momentum $\beta$, Batch Size B.S, consensus step size $\gamma$, $N$ is the number of Nodes in the graph.

**Compute Resources and Reproducibility**

For all the experiments we used a cluster of 8 nodes. 4 of these nodes were equipped with an Intel(R) Xeon(R) Silver 4114 CPU with 93 GB of usable system memory and 3 NVIDIA GeForce GTX 1080 Ti. The other 4 nodes were equipped with an Intel(R) Xeon(R) Silver 4114 CPU with 187GB GB of usable main memory and 4 NVIDIA GeForce GTX 2080 Ti. All the nodes ran CentOS Linux release 7.9.2009 (Core). Regarding packages and versions used for implementation, a detailed requirements file is provided with the source code. In brief, for communication, we use MPI (Gropp et al., 2003) implementation from mpich3.2, for parallel decentralized learning. We used Pytorch framework (Paszke et al., 2019) for automatic differentiation of deep learning models. The source code includes a README file with details on how to reproduce the results. The results reported in the paper we run on seeds 4, 34, and 5. We have tried our best to make the exact runs reproducible, the only requirement being the need to run on the same underlying Nvidia hardware i.e. 1080Ti and 2080Ti. This requirement is due to the application stack details. However, we maintain that when the code is run on different hardware statistically similar results will be obtained.

**Hyper Parameters**

The hyperparameters used for training the models are presented in Table 10. The hyperparameters were tuned using a validation set (10% of the train set). The final accuracy was reported by using these hyperparameters trained on the complete train set. The learning rate is denoted by LR, momentum by $\beta$, batch size by $BS$, number of nodes in the graph by $N$ and consensus step size $\gamma$.

| Dataset | No. of Public Images Exchanged with Neighbor |
|---------|----------------------------------------------|
| CIFAR10 | 15,892 |
| CIFAR100 | 2,559 |
| ImageNette | 21,295 |

Table 11: Average (over 3 seeds and 8 nodes each) number of public images detected as ID (and exchanged) by the OoD detector when the public dataset is TinyImageNet, using an 8 node ring configuration $\alpha = 0.05$

**Computing IDKD overhead**

To calculate IDKD overhead, we consider a training of a 8-node ring on CIFAR10. Assuming a batch size of 32 and the public dataset has 100,000 images (TinyImageNet). On average, the iso iteration epoch is $\approx 262$ for CIFAR100. That is, QG-IDKD at 262 epochs is the same as QG-DGSDm-N 300 epochs (for iso-iteration results, refer to Table 6). Thus if we consider a forward pass compute to be 1x the, backward pass is 2x hence 1 iteration is 3x forward pass. Thus to train a model using QG-DGSDm-N (baseline) for 300 epochs is:

$$cost_{baseline} = 3 \times \text{num iterations} \tag{1}$$

$$= 3 \times \text{num epochs} \times \frac{\text{num train samples}}{\text{num nodes} \times \text{batch size}} \tag{2}$$

$$= 3 \times 300 \times \frac{50,000}{8 \times 32} \tag{3}$$

$$= 175,781.25 \text{ forward passes} \tag{4}$$

To calculate the overhead we use that IDKD is performed every 5 epochs (which is what is used for the results). For the average iso-iteration epochs this results in 3 IDKD exchanges (i.e. 3 = num idkd ex.), thus the overhead is from the forward pass on the public data:

$$cost_{overhead} = \text{num idkd ex.} \times (\text{forward pass on all public images} - \text{forward pass on all selected public images}) \tag{5}$$

This is because we run for iso-iteration for both IDKD and baseline, hence the number of backward passes are the same, the overhead is only from the forward pass on the public dataset. However, forward pass on all public dataset is not wasted since the images identified by the OoD detector can be used in backward pass. This is the term subtracted from Equation 5. From Table 11 for CIFAR10 we see its 15,892. Plugging this into Equation 5 we get:

$$cost_{overhead} = \text{num idkd ex.} \times (\text{num of public image batches} - \text{num of selected public image batches})$$

$$= \text{num idkd ex.} \times \frac{\text{num of public images} - \text{num of public images}}{\text{batch size}}$$

$$= 3 \times \frac{100,000 - 15,892}{32}$$

$$= 3 \times \frac{84,108}{32}$$

$$= 7885.125 \text{ forward passes}$$

Thus the average overhead for CIFAR10 is

$$= \frac{7885.125}{175,781.25} \times 100$$

$$= 4.48\%$$

The best case is 1 IDKD exchange for which the overhead is 1.49% the worst is 10 exchanges for which the overhead is 14.95%.

**Length of Public Dataset**

To investigate the size of the public dataset required for training in CIFAR-10 and CIFAR-100 using IDKD, uniform noise was used as the public dataset. Different sizes of this dataset were tested, including 10000, 20000, 50000, and 100000. These data sets were used to run the QG-IDKD algorithm. The accuracy results of these models were collected and shown in the graph in Figure 5. In this study, an 8-node ring topology with $\alpha = 0.05$ was used. The results shown in Figure 5, suggest that the amount of public dataset needed varies depending on the training set. Generally, both datasets (CIFAR-10 and CIFAR-100) experience improvements with increased data, although gains fade off beyond 50,000.

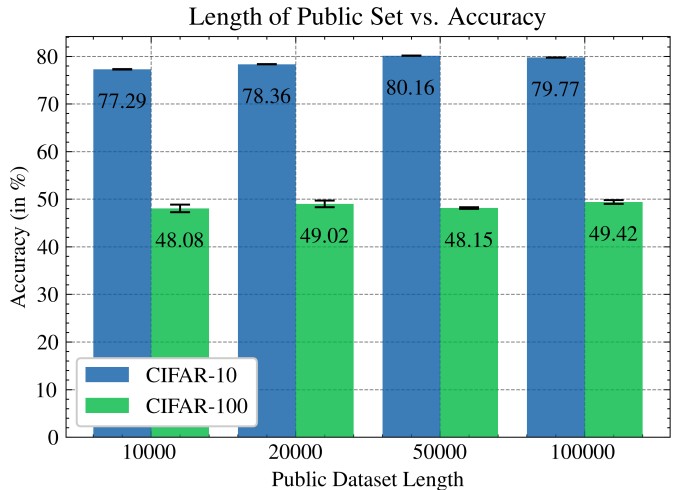

Figure 5: The effect of length of public dataset on the accuracy of QG-IDKD for and 8 node ring topology with $\alpha = 0.05$ for CIFAR-10 and CIFAR-100 datasets.

