# OpenReview forum: "Homogenizing Non-IID Datasets via In-Distribution Knowledge Distillation for Decentralized Learning"
_TMLR — Accepted by TMLR_

### Review · Reviewer_F2Mo · 2024-04-03

**Summary Of Contributions:**

The paper focuses on the challenge of heterogeneity in data distribution in decentralized learning, which leads to slow and unstable global convergence and poor generalization performance. The authors propose a method called In-Distribution Knowledge Distillation (IDKD) to address this challenge by homogenizing the data distribution across nodes without sacrificing privacy. The experiments conducted on multiple image classification datasets and graph topologies demonstrate that the proposed IDKD scheme outperforms traditional knowledge distillation and achieves state-of-the-art generalization performance on heterogeneously distributed data with minimal communication overhead.

**Audience:**

No

**Claims And Evidence:**

Yes

**Requested Changes:**

See my weakness.

**Strengths And Weaknesses:**

Strength:

1. The authors propose a method called In-Distribution Knowledge Distillation (IDKD) to address this challenge by homogenizing the data distribution across nodes without sacrificing privacy

2. To overcome the misalignment between private and public data distribution, an Out-of-Distribution (OoD) detector is introduced at each node to label a subset of the public dataset that maps close to the local training data distribution

3. The experiments conducted on multiple image classification datasets and graph topologies demonstrate that the proposed IDKD scheme outperforms traditional knowledge distillation and achieves state-of-the-art generalization performance on heterogeneously distributed data with minimal communication overhead

Weakness:

1. In the problem setting, it needs to assume there is a public dataset that across every node. I think this is a very strong assumption. The authors need more justification that why this assumption is reasonable and should provide some real-world scenarios.

2. The experimental settings are very unclear. For example, (1) the paper consider different public data size without any analysis such as how many data will be sufficient. (2) The non-i.i.d. setting is unclear.

3. The improvement compared to previous results is very limited while the method in the paper needs a very strong assumption on public data, such as Table 2 and Table 5. And it could be worse than the previous methods.

4. Although the framework is simple, it is only a combination of different approaches directly. I think there should be some novel parts.

---

> ### Author Response · Authors · 2024-05-05
>
> We thank the reviewer for their feedback and for helping improve our paper. We answer the questions from the reviewer below
>
> **Question 1**.	In the problem setting, it needs to assume there is a public dataset that across every node. I think this is a very strong assumption. The authors need more justification that why this assumption is reasonable and should provide some real-world scenarios.
>
> **Authors:** We would like to clarify that while existing public data can be made available at each node, in the absence of such a dataset uniform noise can be used as shown in Table 4 (Table 5 in the revised version). The results in Table 4 show that with IDKD we see performance improvement when using uniform noise as the public dataset. Further, we have added additional results on how much noise data is needed for IDKD (see section 4.2 Impact of Public Dataset Selection and the Appendix - Length of Public Dataset).
>
> **Question 2**. The experimental settings are very unclear. For example, (1) the paper consider different public data size without any analysis such as how many data will be sufficient. (2) The non-i.i.d. setting is unclear.
>
> **Authors:** (1) To clarify this detail, we provide additional results on the number of public samples used (Please see Table 9). Specifically, to address how many public data samples are needed we use an increasing number of uniform-noise samples as the public dataset and plot the corresponding accuracy of an 8 node decentralized training scheme with alpha set to 0.05 on CIFAR-10 and CIFAR-100 datasets. The results have been added to the revised version under the “Length of Public Dataset” subsection of the appendix. The results from this experiment show that as the number of samples of the public dataset increases, the performance increases, up to a certain limit. Further, to provide additional information on other public datasets, we have also included a table with the average number of images exchanged for various datasets, which are the best-case lower bounds on the number of public set images needed (Table 11). Further, we would like to clarify that we have provided the dataset statistics in the appendix and have revised it in the new version to include the statistics for the uniform noise dataset used in the experiments
>
> (2) To clarify the non-IID setup we have re-written the last part of section 4.1. Specifically, we have created a separate section in 4.1 dedicated to explaining the non-IID setup.
>
> **Question 3**. The improvement compared to previous results is very limited while the method in the paper needs a very strong assumption on public data, such as Table 2 and Table 5. And it could be worse than the previous methods.
>
> **Authors:**  We would like to point out that IDKD is targeted at learning under non-iid data distribution. Thus, under larger non-iidness we observe IDKD consistently shows performance improvement (evident from Tables 2 and 5, Tables 2 & 6 in the revised version). Specifically, we always see an improvement when alpha < 0.1 (i.e. more non-iid). To provide a complete picture we chose to also include the result when alpha = 1 (more iid) where the performance of the proposed method is comparable if not better in most cases.

---

### Review · Reviewer_tEFU · 2024-04-07

**Summary Of Contributions:**

This paper presents a data-homogenization approach to a decentralized federated learning with non-IID client data.

The main (high-level) contribution here is the data-homogenization component (via knowledge distillation) which can be added to any existing decentralized deep learning algorithm.

This contribution is motivated from the fact that existing decentralized learning has not considered distilling from an existing public dataset to align and aggregate information across heterogeneous client better.

Thus, the main technical innovation (to substantiate the above high-level contribution) is an OOD-guided knowledge distillation algorithm. Once the decentralized learning algorithm finishes:

1. Each client will use the same public dataset to distill knowledge from its local model, resulting in a (public samples, soft labels) dataset
2. An OOD detector is utilized to determine a subset of the distilled dataset to communicate.

The 2nd step is to ensure that the distillation is focused on the part of the public dataset that is most aligned with the private data.

**Audience:**

Yes

**Claims And Evidence:**

No

**Requested Changes:**

I suggest the following addition:

1. Complete the experiment in Tables 2 & 3. Add more experiment with larger network & more clients.

2. Demonstrate the resilience of the proposed scheme when each node has certain failure probability (see weakness #2) -- will we still have better performance than existing decentralized learning algorithm that this method builds on?

3. Discussing the points I raised in weaknesses #1 and #4

4. Compare with pure engineering solutions to this problem along the direction that I outlined in weakness #3.

**Strengths And Weaknesses:**

Strengths:

The paper is well-written and well-illustrated.

Fig. 2 is very helpful to communicate the high-level idea & motivate the need for using an OOD detector to find the most aligned public subset for each private dataset.

The literature is sufficiently well-covered.

The high-level idea on data homogenization is interesting.

Weakness

I have the following concerns:

First, the paper is not super clear about whether this is about learning a common solution given heterogeneous clients or learning multiple personalized solutions (one per client). Depending on the specific setting, the evaluation method might need to change.

Can the authors provide further clarifications on this?

Second, I think one main motivation for decentralized learning is to remove a choke point of computation and communication. That is, if a node fails, the entire system will still work robustly. However, the paper does not seem to have this evaluation.

Note that, in practical scenarios, even with a server-coordinated federated learning system, only a subset of nodes will participate in the aggregation step. The authors should demonstrate that their data homogenization scheme is robust / resilient against node failure (e.g., setting a drop-out probability for each node)

Third, have the authors considered comparing with a much more simpler, pure engineering solution such as having one node fitting a model on the public dataset and distributing it across the network (sending the model to its neighbors, and their neighbors and so on).

Each client can then fine-tune the public model on its private data. I suspect that this is the might be the most effective way if the setting is to generate personalized solution per node. For example, see https://openreview.net/forum?id=hDDV1lsRV8

Otherwise, if the goal is instead to learn a common solution for all nodes, each client can also distribute its fine-tuned model to its neighbors and hence, to the entire network. This is probably not more expensive than the authors' proposed distillation mechanism.

Fourth, I wonder if the base distributed algorithm (in decentralized system) that the authors leverage will be suitable for federated learning. In my view, one key distinction between distributed & federated learning is that (in addition to data privacy) the latter also wants to minimize the no. of communication rounds, which is achieved via model sharing (rather than gradient sharing).

Is this also the case here? This is not really a show-stopper but it does deserve some discussion (preferably in the main text).

Last, I think the experiment is a bit light. The network size seems to be a bit small (32).
Could the authors run an experiment in larger scale, e.g., more clients (> 100), and with certain probability for node failure (to demonstrate the resilience aspect)?

Furthermore, the experiments in both Tables are not complete. In Table 2, there is no results reported for ImageNette. In Table 3, there is a lack of comparison with most baselines. There is also no result reported for CIFAR datasets.

---

> ### Author Response · Authors · 2024-05-05
>
> We thank the reviewer for their feedback and for identifying experiments that can be used to bolster our results and improve our paper. We answer the questions from the reviewer below
>
>
> **Question 1**. First, the paper is not super clear about whether this is about learning a common solution given heterogeneous clients or learning multiple personalized solutions (one per client). Depending on the specific setting, the evaluation method might need to change. Can the authors provide further clarifications on this?
>
> **Authors:** We would like to clarify that in this paper we focus on global solution and not personalized solution. We have made this clarification in the introduction of the revised paper.
>
> **Question 2**. Second, I think one main motivation for decentralized learning is to remove a choke point of computation and communication. That is, if a node fails, the entire system will still work robustly. However, the paper does not seem to have this evaluation. Note that, in practical scenarios, even with a server-coordinated federated learning system, only a subset of nodes will participate in the aggregation step. The authors should demonstrate that their data homogenization scheme is robust / resilient against node failure (e.g., setting a drop-out probability for each node)
>
> **Authors:** We would like to clarify that decentralized learning has other and possibly more important advantages apart from robustness, such as scalability to large datasets and privacy. Specifically federated systems run into bandwidth limitations at the central server. Further, the central server is a single point of failure for both security and privacy reasons.
> We would also like to add that we considered non-time varying graphs toplogies (as do most baselines). Node failures result in time varying graph topologies. However, for the sake of completeness we provide results on time varying graph topologies with DSGD (known to converge under time varying graphs) as the baseline algorithm. We considered communication failure. At random we introduced communication failures of 1% and 2% (we generate a uniform random number between 0-1 if it is lower than 0.01 or 0.02 we create a failure). The results are shown in Table 4 (in the revised paper), where we see that IDKD results in similar performance gains as the non-failure case.
>
> **Question 3**. Third, have the authors considered comparing with a much more simpler, pure engineering solution such as having one node fitting a model on the public dataset and distributing it across the network (sending the model to its neighbors, and their neighbors and so on). Each client can then fine-tune the public model on its private data. I suspect that this is the might be the most effective way if the setting is to generate personalized solution per node. For example, see https://openreview.net/forum?id=hDDV1lsRV8 Otherwise, if the goal is instead to learn a common solution for all nodes, each client can also distribute its fine-tuned model to its neighbors and hence, to the entire network. This is probably not more expensive than the authors' proposed distillation mechanism.
>
> **Authors:** We would like to clarify that we use unlabeled public dataset, and “one node fitting a model on the public dataset”,  a.k.a  pretraining would require labelled public datasets. Since the goal is to obtain a global solution, a pretrained model would act as better initialization and we believe this does not improve performance.  We use pretraining as the “pure engineering solution”, which we show results in lower performance.  The results for this have been added to Section 4.2 (ablation study, section in the revised paper) where we use a model pretrained on TinyImageNet as initialization and see that generally doing so results in worse performance (See Figure 4 in the revised paper). Regarding “distribute its fine-tuned model to its neighbors” is exactly what DSGD does, which is used as the baseline in Tables 2 and 3. Thus in both cases of using a public data to pretrain, or share the model parameters, IDKD performs better.

---

> ### Author Response · Authors · 2024-05-05
> **Rebuttal Contunied**
>
> **Question 4.** Fourth, I wonder if the base distributed algorithm (in decentralized system) that the authors leverage will be suitable for federated learning. In my view, one key distinction between distributed & federated learning is that (in addition to data privacy) the latter also wants to minimize the no. of communication rounds, which is achieved via model sharing (rather than gradient sharing).
> Is this also the case here? This is not really a show-stopper but it does deserve some discussion (preferably in the main text).
>
> **Authors:** We would like to clarify that a federated system is a special case of a distributed system. For example, a distributed system in a star connection with the node at the center having no data is the same setup as a federated learning system. In the revised paper we provide additional results to show that IDKD works on additional parameter-sharing techniques and thus can be easily adopted in a federated learning setup. We have taken the reviewer feedback under advisement and have added an additional discussion section in the main text in section 4.2 (IDKD and Federated Learning)
>
> **Question 5**. Last, I think the experiment is a bit light. The network size seems to be a bit small (32). Could the authors run an experiment in larger scale, e.g., more clients (> 100), and with certain probability for node failure (to demonstrate the resilience aspect)?
>
> **Authors:** Please note we provide results for a node size of 64 in Table 7 (Table 8 in the revised paper). More nodes of 100 or greater spreads the data too thin since each client for CIFAR10/CIFAR100 has less than 500 samples each. We would like to point out that fewer samples per node results in more noisy results which are less interpretable (seen to some extent on ImageNette already i.e. see Table 3 in the revised paper where baseline ImageNette results have the largest performance variance on average i.e. std numbers are large). Finally, while federated learning papers perform large node analysis, 64 is a large number of nodes in the “decentralized community” as seen in the references of QG-DSGD-mN (32 nodes) and Relay (16 nodes), thus we believe the results presented in the paper conform to standard practice.
>
> Regarding node failures, we provide detailed analysis under a new subsection “Node Failures (Time-Varying Graphs):”. This section in the revised paper presents findings using DSGD and SGP both of which are known to converge under time-varying graphs (i.e. node failures) and show improvement even under node failures.
>
> **Question 6**. Furthermore, the experiments in both Tables are not complete. In Table 2, there is no results reported for ImageNette. In Table 3, there is a lack of comparison with most baselines. There is also no result reported for CIFAR datasets.
>
> **Authors:** We have revised the paper to include more baselines and provided additional results to be more comprehensive.
>
> Requested Changes:
>
> 1.	Complete the experiment in Tables 2 & 3. Add more experiment with larger network & more clients.
>
> **Authors:** We have completed the tables with additional results and for node failure, please note contemporary decentralized methods do not consider node failures (due to node failure being classed under time-varying graphs which complicates convergence results) however for the sake of completion we have provided the results in Table 4 (of the revised paper). As mentioned previously. we provide results for 64 nodes in Table 8.
>
> 2.	Demonstrate the resilience of the proposed scheme when each node has certain failure probability (see weakness #2) -- will we still have better performance than existing decentralized learning algorithm that this method builds on?
>
> **Authors:** We provide these results in Table 4 (of the revised paper) where we consider 1% and 2% communication failure rates. Which are modeled to occur randomly with uniform probability during any communication. We show we still see improved performance on decentralized learning algorithms.
>
> 3.	Discussing the points I raised in weaknesses #1 and #4
>
> **Authors:** We have added clarifications in the paper regarding #1 and #4 by adding clarifying statements in the introduction and adding a discussion section on federated learning.
>
> 4.	Compare with pure engineering solutions to this problem along the direction that I outlined in weakness #3.
>
> **Authors:** We have added additional results to bolster the results with “pure engineering solutions” which are included in the revised version of the paper.

---

### Review · Reviewer_7rgJ · 2024-04-28

**Summary Of Contributions:**

The paper introduces In-Distribution Knowledge Distillation (IDKD), a novel method designed to improve decentralized learning on non-IID datasets. It addresses data heterogeneity across nodes by using a public dataset for knowledge distillation without compromising privacy. By incorporating an Out-of-Distribution (OoD) detector, IDKD selectively distills knowledge based on data alignment, thus overcoming the common pitfalls of traditional knowledge distillation methods. The experimental results show that IDKD outperforms existing methods in terms of generalization performance on diverse image classification tasks and network configurations, achieving this with minimal communication overhead. This approach provides an efficient solution for homogenizing data distribution in decentralized environments while maintaining privacy.

**Audience:**

No

**Broader Impact Concerns:**

Not involved.

**Claims And Evidence:**

No

**Requested Changes:**

1. To better understand the efficacy and contribution of each individual component of your proposed IDKD method, an ablation study is essential. This study should detail how each component (e.g., the OOD detector and the specific distillation strategy) contributes to the overall performance improvements.
2. Your manuscript currently lacks a detailed analysis of the computational costs associated with implementing IDKD in decentralized settings.

**Strengths And Weaknesses:**

Strengths
1. Effective Handling of Data Heterogeneity: The paper successfully tackles the issue of heterogeneous data distributions across nodes—a common obstacle in decentralized systems.
2. Minimal Communication Overhead: Another significant strength of the paper is its achievement in reducing the communication overhead typically associated with decentralized learning.

Weaknesses
1. Lack of Ablation Study: One significant weakness in the paper is the absence of an ablation study.
2. Lack of Computation Cost Analysis: The paper does not provide a detailed analysis of the computational costs associated with the proposed IDKD method.

---

> ### Author Response · Authors · 2024-05-05
>
> We thank the reviewer for their feedback and for helping improve our paper. We answer the questions from the reviewer below
>
> **Question 1.** To better understand the efficacy and contribution of each individual component of your proposed IDKD method, an ablation study is essential. This study should detail how each component (e.g., the OOD detector and the specific distillation strategy) contributes to the overall performance improvements.
>
> **Authors:** We would like to clarify that “QG-DSGDm-N + KD” in Table 2 is effectively an ablation of the OoD detector, hence the results in Table 2 serve this purpose. However, we agree with the reviewer and have added a dedicated section under 4.2 for an ablation study to show the effect of KD and OoD detector separately. Further, we also added an additional result to compare the performance of pretraining using the public dataset. All of the results are visualized in Figure 4 (in the revised paper) and show the improvement with the addition of each component of IDKD (i.e. KD and OoD detector).
>
> **Question 2.** Your manuscript currently lacks a detailed analysis of the computational costs associated with implementing IDKD in decentralized settings.
>
> **Authors:** We have updated the computational cost section of the paper. A detailed breakdown of how the cost was calculated is also provided in the appendix of the revised paper, while doing so we realized we had previously failed to account for the forward passes of the public dataset in our calculation, including that we find that the compute overhead for a public dataset of size 100k is between ~1.5 - 15%  (based on how many number of iterations, which is based on the number of public samples detected by the OoD detector as in-distribution) and 4.5% on average. The overheads are relative to the baseline training algorithm.

---

### Decision · Action_Editor_BYgM · 2024-06-14

**Recommendation:** Accept as is

**Comment:**

All reviewers agreed that the work should be accepted, and basically seem to be in agreement that while there are experiments, settings, and comparisons that could strengthen the work (e.g. larger scales, more challenging communication settings) and that the reliance on a public dataset is potentially a drawback (though as the authors note, their method works even when initializing the public dataset to be purely noisy), the paper gives enough evidence towards its stated accomplishments that it warrants acceptance.

Reviewers specifically highlighted the breadth of experimental comparisons (to other methods, to other settings, in various topologies and on various datasets) as a strength that outweighed any reservations they had. Thus, I recommend acceptance as is.

**Audience:**

Yes. As Reviewer 7rgJ and F2Mo, the idea of studying things at the intersection of data heterogeneity and decentralized learning is currently a very active topic. As Reviewer tEFU notes (quite elegantly) in their recommendation, this work should be viewed as orthogonal but potentially synergistic with federated learning (which is an extremely active field that some of TMLR's audience would be interested in). While the connections to federated learning are not 100% certain (as tEFU notes) the results for decentralized learning are convincing enough that readers would likely be interested in understanding the pertinence to federated learning, potentially building on the method conceptually.

**Claims And Evidence:**

The reviewers generally agree that the paper is well-written, and provides a variety of interesting experimental evidence towards the efficacy of their methods. While reviewer tEFU cautions that some of the empirical studies are a bit light (especially when considering the actual communication setting of interest, e.g. whether there are communication failures versus partial participation of clients) they still recommend acceptance based on the thoroughness of experiments in the specific settings studied by the authors.